# Interfacial Spin Manipulation of Nickel-Quinonoid Complex Adsorbed on Co(001) Substrate

**Indukuru Ramesh Reddy [1], Peter M. Oppeneer [2] and Kartick Tarafder [1,*]**

[1] Department of Physics, National Institute of Technology Karnataka, Srinivasnagar, Surathkal, Mangalore 575025, India; reddy.indukururameshreddy@gmail.com
[2] Department of Physics and Astronomy, Uppsala University, Box 516, S-75120 Uppsala, Sweden; peter.oppeneer@physics.uu.se
[*] Correspondence: karticktarafder@gmail.com; Tel.: +91-824-2473927

**Abstract:** We studied the structural, electronic, and magnetic properties of a recently synthesized Ni(II)-quinonoid complex upon adsorption on a magnetic Co(001) substrate. Our density functional theory $+U$ (DFT+$U$) calculations predict that the molecule undergoes a spin-state switching from low-spin $S = 0$ in the gas phase to high-spin $S \approx 1$ when adsorbed on the Co(001) surface. A strong covalent interaction of the quinonoid rings and surface atoms leads to an increase of the Ni–O(N) bond lengths in the chemisorbed molecule that support the spin-state switching. Our DFT+$U$ calculations show that the molecule is ferromagnetically coupled to the substrate. The Co surface–Ni center exchange mechanism was carefully investigated. We identified an indirect exchange interaction via the quinonoid ligands that stabilizes the molecule's spin moment in ferromagnetic alignment with the Co surface magnetization.

**Keywords:** spin-state switching; interfacial spin-manipulation; DFT+$U$ theory

## 1. Introduction

Molecular spintronics has gained enormous attention in the past two decades because it holds promise for advanced multifunctional spin devices [1,2]. Especially spin-bearing molecules with one or more transition metal (TM) ions having a partially filled $d$ shell with $d^4$ to $d^7$ configurations have drawn special attention as for many such complexes the spin properties can be manipulated externally. For spin-crossover complexes, it has been shown that it is possible to switch reversibly between different spin states via an external perturbation such as pressure, temperature, light irradiation, electric or magnetic fields [3,4]. Many magnetic molecules have been synthesized and intensively investigated both theoretically and experimentally in the past years [5–8]. The progress made recently in the application of spin-crossover complexes in functional devices has been reviewed lately [9,10]. The understanding of the underlying electronic and magnetic interactions on the single molecule level is, however, still sparse. Efficient and deterministic spin detection, manipulation and transportation in organic materials are yet to be achieved, properties that form an integral part of any advanced spin device.

Growing molecules on magnetic as well as inert surfaces with atomic precision is one of the approaches to develop such devices. Immobilized magnetic metal–organic molecules with tailorable magnetic properties can be deposited on solid substrates to construct building blocks for molecular spintronic devices [11,12]. In this process, the molecules exhibit magnetic stability at higher temperatures due to the substrate induced magnetic exchange coupling and/or substrate induced magnetic anisotropy [13–15]. Moreover, new magnetochemical properties arise due to the molecule–substrate interaction that can also offer an advantage over traditional spintronics [16–20].

It has been observed that magnetic molecules reacting on a metallic surface may lose their magnetic properties by coupling with the electrodes, even if they remain intact [21]. Hence, a proper understanding of molecule–substrate interactions is essential and needs to be achieved well before its technological application in data storage, sensor, quantum computing, or other single molecular spintronic device applications [22].

An isolated molecule that is at a relatively large distance from the surface/electrode or in the gas phase has discrete molecular energy levels. The bonding interactions may occur at the interface when molecules are brought near to the surface and adsorbed on it. The interaction can vary from weak (in physisorption) to strong bonding interactions (chemisorption) depending on the physical and chemical nature of the molecule and surface. The electronic structure of adsorbed molecules may change considerably. In physisorption, the molecule–substrate separation is relatively large, interaction is relatively weak, and the discrete molecular electronic levels are weakly broadened. On the other hand, electronic levels of chemisorbed adsorbates are significantly changed. Atoms of the adsorbed molecule in proximity with surface atoms form direct chemical bonds with surface atoms, which results in a much stronger substrate–molecule interaction. Particularly, in the case of magnetic substrate, new spin-resolved interface electronic states may appear near the Fermi energy due to the $\pi - d$ hybridization [23,24] that may completely change the electronic structure of the molecule and also affect the molecular geometry. To study and understand the chemical and magnetic exchange interactions between the adsorbed metal–organic molecules and magnetic substrates is therefore a fundamental focus. Such study may reveal an effective spin manipulation mechanism of the adsorbed molecule. The chemical environment between the adsorbed molecule and substrate determines the type and strength of magnetic exchange interactions which can be controlled externally by changing chemical environment [25,26]. Efficient chemical modification can lead to a significant change in $\pi - d$ hybridization across the interface and provides a potential path for molecular magnetism that is stable at room temperature ferromagnetism [13,23]. Hence, the interface magnetic properties that arises due to interface chemistry of molecule and surface, demand a careful study with the help of advanced computational and sophisticated experimental tools (XMCD and SP-STM) [27] to provide an insight into the spin-interface chemistry [28,29].

In general, non-planar molecules would bind weakly to a metal surface due to steric hindrance, whereas planar molecules could bind more strongly or self-assemble on the metal surfaces. Therefore, planar molecules are suitable candidates to design molecular devices by depositing them on matching metal surfaces. A number of theoretical as well as experimental observations have been reported on the magnetic interactions of transition-metal porphyrins (TM-P) [13,23], transition-metal phthalocyanine (TM-Pc) [12,30,31], and cyclohexane-based organic molecules [32] with metal substrates. It has been observed that planar TM-P and TM-Pc molecules are very promising as they show a stable magnetic behavior due to exchange coupling to the magnetic substrate. Moreover, their magnetic properties can be tailored by engineering the spin states of the metal center using on-surface magnetochemistry, i.e., chemical perturbation by coordination in the free ligand position or through interlayer modified exchange interaction [25,29,33–36]. Spin-state switching is also observed for a planar metal–organic molecule due to stretching the metal–ligand distance [37], similar to the bond-length induced switching of spin-crossover molecules [3]. While this focuses primarily on bond-length changes, low-spin to high-spin state switching in addition brings about thermodynamic changes such as entropy increase, which could play a role in the adsorption process.

Recently, Kar et al. designed a methanol-triggered vapochromism and spin-switching in a Ni(II)-quinonoid (NiQ) complex [Ni(HL)$_2$, H$_2$L = 4-methylamino-6-methyliminio-3- oxocyclohexa-1, 4-dien-1-olate], which is a planar molecule. They showed that the addition of methanol (MeOH) to the molecule leads to a magnetochemical reaction, in which the central Ni atom of the molecule changes it co-ordination from square-planar to octahedral by axial ligation of two MeOH groups at the Ni site, accompanied by a temperature-robust spin transition from $S = 0$ to $S = 1$ [38]. This observation suggests that the Ni(II)-quinonoid could be a potential candidate material for application in spintronics

devices. However, for applications, a detailed understanding of the molecular adsorption mechanism on a metal surface is essential and, in addition, a major question is if magnetic interface interactions are sufficient to trigger the spin-switching.

The main objective of the present investigation was therefore to study the geometric and electronic structure of the multifunctional NiQ molecule when adsorbed on a Co(001) substrate, the emergent magnetic interactions at the spin-interface, and, specifically, whether an on-surface spin-state transition can be achieved. Thereto, we carried out a detailed theoretical investigation using density functional $+U$ theory to unravel the nature of adsorption, electronic structure, and the magnetic state of the adsorbed NiQ molecule. Our investigation predicts that the NiQ molecule undergoes an on-surface spin-state transition to a high spin state when adsorbed on a Co(001) surface.

## 2. Computational Methodology

We employed density functional theory (DFT) calculations to study the magnetic interaction of a NiQ molecule with a ferromagnetic cobalt surface. The calculations were performed using the plane wave, pseudo-potential method, as implemented in Vienna Ab-initio Simulation Package (VASP) [39] with projector augmented plane wave (PAW) potential [40]. For the exchange-correlation function, the generalized gradient approximation (GGA) was used with the Perdew–Burke–Ernzerhof parameterization [41]. We further used the DFT+$U$ technique to capture the strong electron–electron correlation effect that exists in the partially filled $3d$ shell of Ni and the missing correlation effect beyond the plain GGA. This technique was proven to be very promising to achieve the precise spin state of the molecule [24,42–44]. The on-site Coulomb and exchange parameters $U$ and $J$ used in this study were chosen as 5.0 eV and 1.0 eV for Ni atom. These values of the $U$ and $J$ parameters have previously been found to provide an accurate description of the low and high spin states of bi-stable metalorganic molecules [44–47]. A $3 \times 3 \times 1$ Monkhorst–Pack $k$-point grid was used for reciprocal space sampling. A sufficiently large plane-wave kinetic energy cutoff 500 eV was considered to obtain the desire accuracy. The convergence criterion was set to be $10^{-5}$ eV for the self-consistent electronic minimization. We modeled the Ni(II)-quinonoid molecule on top of three atomic Co layers within a large super-cell of the Co(001) magnetic substrate (we used $8 \times 5$ lateral supercell) and applied boundary conditions to maintain the periodicity of the surface. We optimized the geometry of the molecule on the Co surface by relaxing the atoms' positions using the conjugate gradient method until all residual inter-atomic forces were minimized up to 0.01 eV/Å. Two bottom layers of Co atoms of the substrate were kept fixed during the optimization process to minimize the computational effort.

## 3. Results and Discussion

### 3.1. Optimized Geometry

We modeled the square-planar NiQ molecule structure according to the reported data [38]. To start with, we optimized the geometry of the isolated NiQ molecule in the gas phase. The optimized structure of NiQ is shown in Figure 1. The molecule remains planar with average Ni–O(N) bond lengths of the molecule being 1.868 Å, which is close to the experimental value (1.866 Å).

Next, we placed the optimized NiQ molecule on the Co(001) substrate in two different configurations, namely the HOLLOW and TOP adsorption sites. In the HOLLOW configuration, the molecule adsorbed on the Co(001) surface in such a way that the central Ni atom of the molecule is placed on top of a second layer Co atom (i.e., hollow site) of the Co(001) substrate (see Figure 2). The optimized distance between the surface and Ni atom is 1.778 Å. In this configuration, the Ni–O and Ni–N bond lengths have been stretched in such a way that the O and N atoms of the molecule come closer to surface Co atoms and form direct chemical bonds with the surface. The average Ni–O(N) bond length is increased by 0.204 Å compared to its gas phase structure.

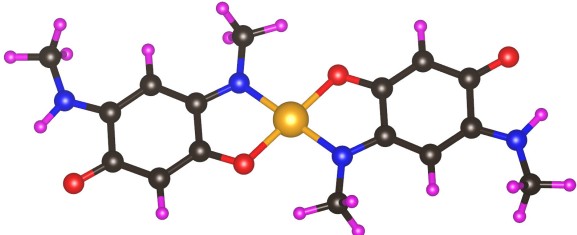

**Figure 1.** The optimized structure of the nickel quinonoid molecule. Ni, O, N, C, and H atoms are represented through the orange, red, blue, black, and magenta colored balls, respectively.

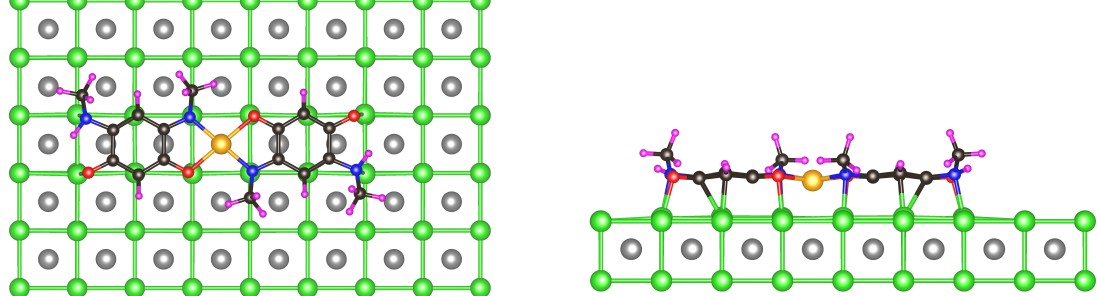

**Figure 2.** Top and side view of the chemisorbed NiQ molecule on the surface of Co(001), in the HOLLOW configuration. The NiQ atoms are represented in the same way as in Figure 1, and the Co atoms are represented by green and gray colored spheres. To make the Co positions in the second layer (hollow site) clearer, these Co atoms are represented with gray color and their bonds with other layer Co atoms are not shown.

In TOP configuration (see Figure 3), the molecule is placed on the Co (001) surface such that the Ni atom sits exactly on top of a surface Co atom. The optimized molecule substrate separation in this configuration is 2.4864 Å. On optimization, the ligands connected to the Ni atom of the molecule are impetus to stretch and orient themselves in such a way that the central O atoms of the molecule, which were initially at the hollow site, move toward the top site. Therefore, the bond length between Ni and O is increased approximately by 0.26 Å compared to the gas phase, whereas the Ni–N bond length does not change significantly. This is mainly because the nearest C atoms, which were already bonded with surface Co atoms, prevent the further stretching of Ni–N bonds. The average bond length is increased by 0.128 Å. Our calculations show that the HOLLOW configuration is energetically more favorable, whereas the molecule is structurally distorted more in the TOP configuration. The magnetic moment of the Ni atom, Ni–O(N) bond length in gas phase, and in different chemisorbed configurations are shown in Table 1.

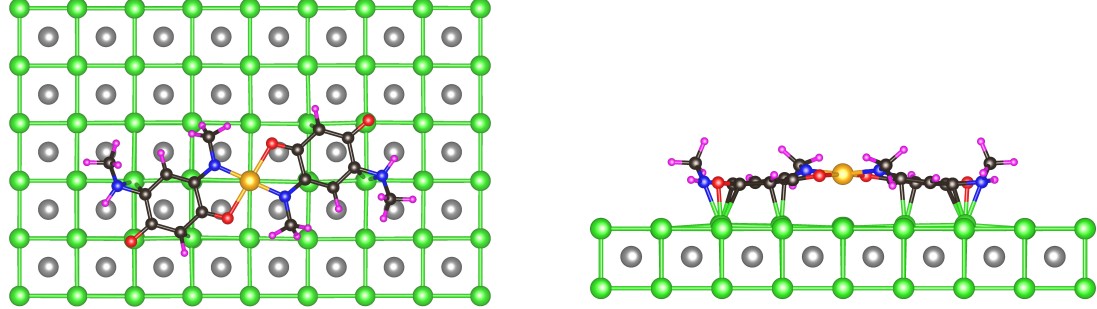

**Figure 3.** Top and side view of the chemisorbed NiQ molecule on the surface of Co(001), in the TOP configuration. The color code of the atoms is done in the same way as in Figure 2.

**Table 1.** The Ni–O and Ni–N bond lengths (in Å), magnetic moment on Ni, and total molecular moment (in $\mu_B$) of NiQ in gas phase, and chemisorbed in HOLLOW and TOP configurations.

|  | Gas phase | HOLLOW | TOP |
| --- | --- | --- | --- |
| Ni–O1 | 1.8562 | 2.1151 | 2.1156 |
| Ni–O2 | 1.8558 | 2.1158 | 2.1166 |
| Ni–N1 | 1.8782 | 2.0290 | 1.8778 |
| Ni–N2 | 1.8799 | 2.0295 | 1.8778 |
| Average | 1.8683 | 2.0724 | 1.9967 |
| Magnetic Ni moment | 0.000 | 1.178 | 1.133 |
| Magnetic moment | 0.000 | 1.561 | 1.337 |

Our calculations predict a zero moment ($S = 0$) for the gas-phase NiQ molecule, consistent with experiments [38]. We note, with regard to the obtained $S = 0$ state, that we verified that the zero-moment property is robust against variations of the Coulomb $U$ value between 2 and 8 eV. The molecular spin moment changes when the molecule is chemisorbed on the Co(001) surface, to 1.561 and 1.337 $\mu_B$ for HOLLOW and TOP site, respectively. Thus, an on-surface spin-state switching to a high-spin state can be induced by adsorption of NiQ on the Co substrate. Compared to the reported magnetochemical reaction [38] in which two MeOH groups bond to Ni in the axial ligand positions, in the on-surface reaction, the molecule bonds one-sided to the top-layer surface atoms. The spin moment on the molecule is less than what would be expected for the high-spin $S = 1$ state. This is however not unexpected due to the extended hybridization of the quinonoid atoms to the surface atoms and the structural distortion.

The physisorption of this class of molecules is also possible on the Co surface [42]. Therefore, we looked for such physisorption of NiQ on Co surface. The distance between the NiQ molecule and Co(001) surface was successively changed in both HOLLOW and TOP configurations and the total energy of the systems in each situation calculated. The computed energy versus distance curve for the HOLLOW site absorption is shown in Figure 4. The single energy minima at 1.8 Å clearly indicates that only chemisorption of the molecule on Co(001) is possible. A weak physisorption of the molecule was observed at a distance 3.5 Å in TOP configuration, however the calculated energy barrier was very small and the magnetic interaction was found to be negligible in this configuration. The details of geometric structure and magnetic behavior in physisorption in TOP geometry are given in Appendix A.

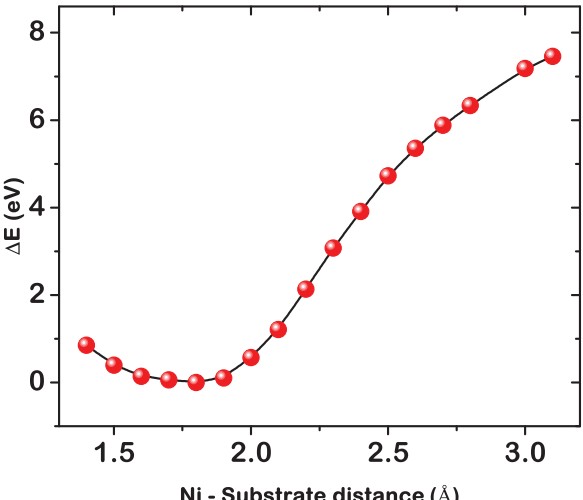

**Figure 4.** Calculated relative total energy (ΔE) versus the molecule–surface distance. The total-energy curve clearly displays its energy minima at 1.8 Å, which corresponds to chemisorption of the NiQ molecule on the Co(001) substrate in HOLLOW configuration.

### 3.2. Electronic Structure and Magnetic Properties

The distorted molecular geometry in chemisorption leads to a significant change in the electronic energy levels and their spin occupation, which further affects the magnetic properties of the molecule. In the square planar Ni(II)-quinonoid molecule with $d^8$ configuration, the four ligands (two oxygen and two nitrogen atoms) are bonded to the central Ni atom in the basal $xy-$plane and provide a crystal-field effect on the Ni $3d$-orbitals. The Coulomb repulsion along with strong $\sigma$-type hybridization leads to a strong ligand field (LF) that splits the $d$-orbitals of Ni in degenerate $t_{2g}$ and $e_g$ levels. In the absence of an axial co-ordination, one of the $e_g$ orbitals, i.e., $d_{z^2}$ is relatively lower in energy compare to the other, $d_{x^2-y^2}$. A stronger LF effect is imposed on the orbitals when the Ni–O and Ni–N bond lengths are shorter due to the strong hybridization and Coulomb repulsion. As a result, all $d$-electrons occupy the lower, LF-splitting dominated energy levels and yield a low-spin state ($S = 0$) in the gas phase, where all $d$-orbitals are completely occupied, except $d_{x^2-y^2}$ which is completely empty (see Figure 5a).

To investigate the charge transfer between molecule and substrate, we have calculated the Bader charges for the chemisorbed NiQ molecule in HOLLOW and TOP configurations. The molecule gains 2.267 and 1.959 electron charges from the Co surface in the HOLLOW and TOP configuration, respectively. The Bader charge on the Ni atom increases however only by 0.169 and 0.090, respectively.

In the case of molecular chemisorption, electrons in the outer-plane $\pi$-orbitals of the quinonoid strongly interact with the surface state electrons and form chemical bonds between the molecule and surface atoms. As a result, the Ni–O(N) bond lengths are stretched, which leads to a significant reduction of the LF effect and subsequently reduces the $d_{x^2-y^2}$ orbital energy. In this situation, the intra-atomic exchange interaction becomes important and the electrons occupy the Ni-$3d$ orbitals according to Hund's rule. Hence, the $d_{x^2-y^2}$ orbital level, which was completely unoccupied before, is now partially occupied (see Figure 5b,c). This explains the change in the spin state of a chemisorbed NiQ molecule governed by the shift in the energy level of the $d_{x^2-y^2}$ orbital, as shown in Figure 5.

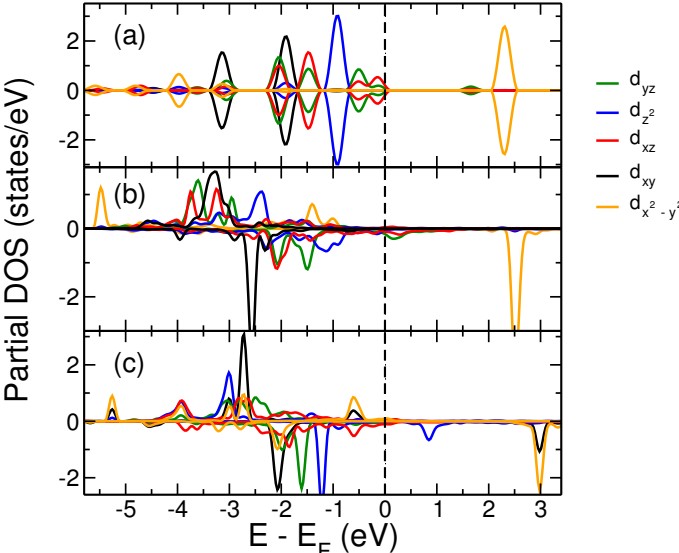

**Figure 5.** $d$-orbital projected DOS of Ni atom in: (**a**) gas phase NiQ; (**b**) chemisorbed in HOLLOW configuration; and (**c**) chemisorbed in TOP configuration. Spin-up and spin-down DOS are shown by positive and negative values, respectively.

The magnetic exchange interaction energies $E^{ex}$ in different chemisorption configurations of the molecule are calculated using $E^{ex} = E^{AFM} - E^{FM}$, where $E^{AFM}$ and $E^{FM}$ are the total energies of anti-parallel and parallel spin alignment between the central Ni atom of the molecule and substrate Co atoms, respectively. Our calculation shows that the molecule is ferromagnetically coupled with the substrate with coupling energy of 68 and 88 meV for HOLLOW and TOP configurations, respectively.

To understand the spin coupling mechanism, we carefully investigated the atom projected density of states (PDOS) of the systems. In the HOLLOW configuration, the Ni-ligating O and N atoms are located on top of the surface Co atoms and form chemical bonds with the substrate Co atoms underneath (Figure 7). We observed that there are overlapping peaks of Co, N, and O projected densities of states marked with vertical arrows in the Figure 6a, indicating strong hybridization between out-of-plane orbitals ($d_{z^2}$, $d_{xz}$, $d_{yz}$, and $p_z$) of Co, O, and N atoms. The total charge density plot, shown in Figure 7 (left) as light green hypersurface, also confirms that the orbital overlapping is only between surface Co and molecular ligand's orbitals as there is no direct charge density overlap between the central Ni atom of the molecule and the Co surface atoms. This clearly indicates that the molecule interacts with the substrate only through its ligands, which in turn strongly favors magnetic coupling through the indirect super exchange interaction.

We computed a total 1.561 $\mu_B$ spin moment on the molecule in HOLLOW position, whereas 1.178 $\mu_B$ spin moment is located on the central Ni atoms (Table 1). A small spin-polarization is also observed on the ligand atoms, which further indicates that the exchange coupling path between the magnetic center of the molecule and the spins of Co surface atoms goes through the molecular ligands. To obtain more insight in the nature and path of the magnetic exchange interaction between magnetic centers, we computed the *partial* magnetization density of the system, shown in Figure 8, at the energy region where the hybridization between atomic orbitals of Co surface Co and molecular orbitals from N, O, and Ni are dominant, which is predicted from the atom projected DOS (see the overlapping peaks in Figure 6a). It is evident from the magnetization density plot that the magnetic exchange interaction between Ni and surface Co atoms is mainly mediated through ligands of the molecule. The negative magnetic moments on O and N atoms, shown through the light blue isosurface in this figure, indicate a 90° ferromagnetic indirect exchange mechanism with the Kanamori–Goodenough superexchange rule. A very weak direct exchange interaction could also be possible when one considers some of the noticeable PDOS peaks of Ni and Co atoms that have overlap, as indicated with vertical bars in the atom projected DOS shown in Figure 6a. However, we did not find any direct charge or spin-density overlap between Ni and Co atoms in the charge density and magnetization plots.

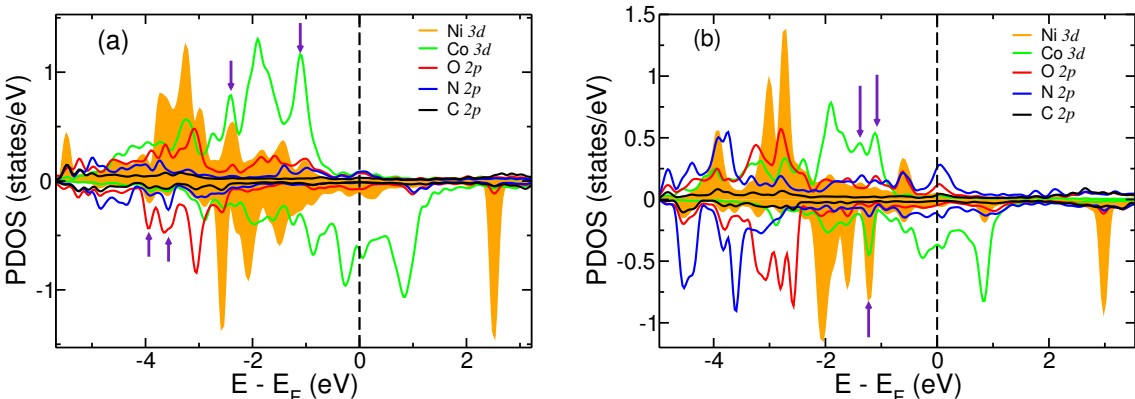

**Figure 6.** Atom-projected partial DOS of chemisorbed NiQ molecule on Co(001): (**a**) NiQ adsorbed in HOLLOW configuration; and (**b**) NiQ adsorbed in TOP configuration. Spin-up and spin-down DOS are shown by positive and negative values, respectively.

In the case of chemisorption of the molecule in the TOP configuration, the direct O and N atoms bonding to the Ni atom are in the hollow position with respect to the surface atoms and do not directly bond to the surface Co atoms; however, there are other atoms of the quinonoid ligand that chemically bond to the surface. The total charge density plot in Figure 7 (right) clearly indicates the non-overlapping nature of the orbital wave functions between the central Ni atom and the surface. However, quinonoid ligand atoms of the molecule strongly interact with the surface Co atoms. There are overlapping peaks present in the atom-projected PDOS plot (Figure 6b) marked with vertical arrows, which indicates the strong hybridization between out-of-plane orbitals ($d_{z^2}$) of surface

Co with O and N $p-$orbitals. The magnetization density calculated for the TOP configuration is shown in Figure 9. A small negative magnetization density is present on the ligand atoms, as shown by the light blue isosurface that indicates an indirect exchange interaction between the Ni spin center and the magnetic Co surface atoms, mediated through the ligand.

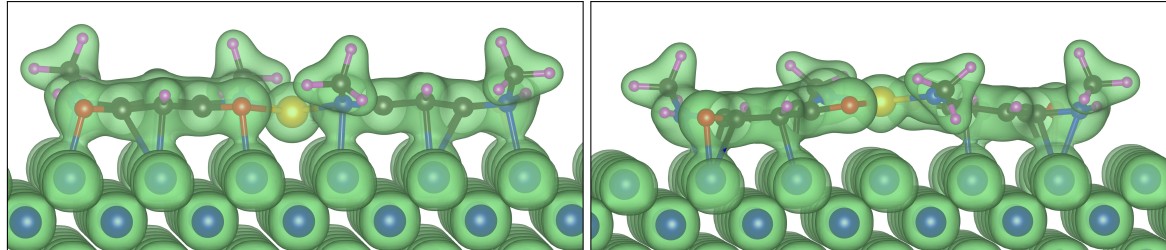

**Figure 7.** Computed charge density (depicted by the light green hypersurface) of NiQ: (**left**) in HOLLOW configuration; and (**right**) in TOP configuration. The charge densities reveal that hybridization between the surface and molecule occurs only through the quinonoid ligands.

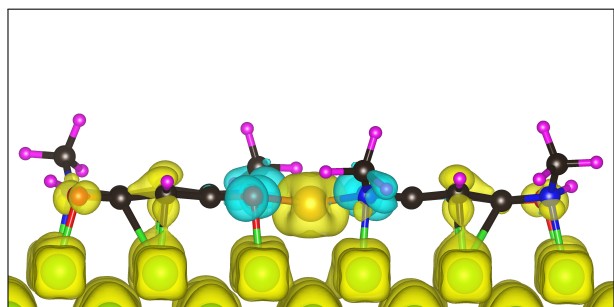

**Figure 8.** Partial magnetization density computed for an energy window indicated with two arrow marks in the spin down channel of the PDOS plot for the HOLLOW configuration (Figure 6a). The magnetization density clearly shows the ferromagnetic coupling between molecule and surface, and also that an indirect superexchange mechanism is more dominant than the direct exchange interaction. The bright yellow and light blue hypersurfaces depict the positive and negative magnetization densities, respectively.

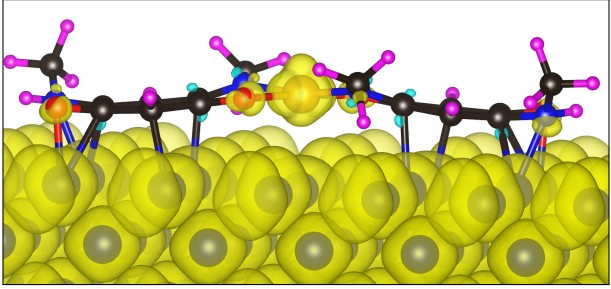

**Figure 9.** Computed total magnetization density of NiQ molecule chemisorbed in the TOP configuration. The magnetization density on ligand N and C atoms points to a ferromagnetic coupling between the molecule and substrate through a direct exchange path. The bright yellow and light blue hypersurfaces depict the positive and negative magnetization densities, respectively.

## 4. Conclusions

To conclude, we have demonstrated that the adsorption of the NiQ molecule on a Co(001) surface leads to a spin crossover from $S = 0$ in the gas phase to a nearly high-spin $S = 1$ state when chemisorbed on the Co surface. Chemisorption at a HOLLOW site is found to be energetically most favorable. The covalent interactions between the molecule's ligands and surface lead to a stretching of the Ni–O and Ni–N bond lengths that lead to the spin-crossover of the non-magnetic NiQ into a

spin bearing molecule. An indirect superexchange path through the molecular ligand stabilizes the Ni spin of the molecule parallel to the substrate magnetization direction. In view of the on-going efforts to realize bi-stable molecules with reversible spin state that are atomically precise anchored to substrates, a future research direction is to explore the possibility of achieving spin-switching on non-magnetic metallic substrates. Our investigation not only sheds light on the interactions of a NiQ molecule with a magnetic Co(001) substrate, but also suggests a possible pathway to explore planar metal-quinonoid molecules as spin-bearing interface molecule in future spintronic devices.

**Author Contributions:** K.T. conceived of the presented idea and designed the study. I.R.R. performed the computations. K.T. and P.M.O. supervised this work. All authors discussed the results and contributed to the final manuscript.

**Funding:** We gratefully acknowledge financial support from the Indo-Swedish Research Collaboration, funded through the Swedish Research Council (VR) and the Department of Science and Technology, India (DST-SERB, Project No. SB/FTP/PS-032/2014), and the Knut and Alice Wallenberg Foundation (Contract No. 2015.0060).

**Acknowledgments:** We acknowledge computer time received from the Swedish National Infrastructure for Computing (SNIC).

**Conflicts of Interest:** The authors declare no conflict of interest.

## Abbreviations

The following abbreviations are used in this manuscript:

| | |
|---|---|
| DFT | Density Functional Theory |
| DFT+$U$ | Density Functional Theory +$U$ |
| XMCD | X-ray magnetic circular dichroism |
| SP-STM | Spin-polarized scanning tunneling microscopy |
| TM | Transition metal |
| TM-P | transition-metal porphyrin |
| TM-Pc | Transition-metal phthalocyanine |
| NiQ | Ni(II)-quinonoid |
| MeOH | Methanol |
| VASP | Vienna Ab-initio Simulation Package |
| PAW | Projector augmented plane wave |
| GGA | Generalized gradient approximation |
| LF | Ligand field |
| DOS | Density of states |
| PDOS | Partial density of states |

## Appendix A. Physisorption of NiQ on Co(001)

We studied theoretically the possible physisorption of the NiQ molecule on Co(001) in the TOP configuration. The total energy of the molecule in TOP configuration versus the molecule–substrate distance is shown in Figure A1. The shallow energy minima at 3.5 Å clearly indicates that physisorption of the molecule is possible. We note however that we have not included Van der Waals interactions (see, e.g., [48,49]) in the present calculation, but these would be required for an accurate description of the physisorption. The shallow minimum obtained at 3.5 Å is hence mainly driven by higher-order electrostatic interactions [42]. The optimized geometry of physisorbed NiQ on the Co surface is shown in Figure A1 (right). In the physisorption state, the NiQ molecule preserves mostly its gas-phase properties. The Ni–O and Ni–N bond lengths, the average Ni–O(N) bond length (1.8764 Å), and the magnetic moment on the Ni atom are very close to those of the gas-phase molecule. The computed charge and magnetization densities are shown in Figure A2. The charge density plot shows that the there is no orbital overlap between the molecule and the substrate, and the magnetization density plot exemplifies the non-magnetic nature of the physisorbed molecule on substrate.

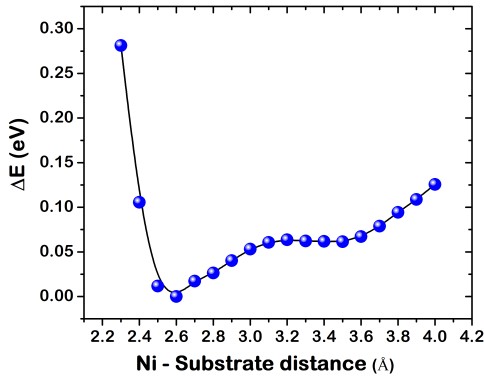 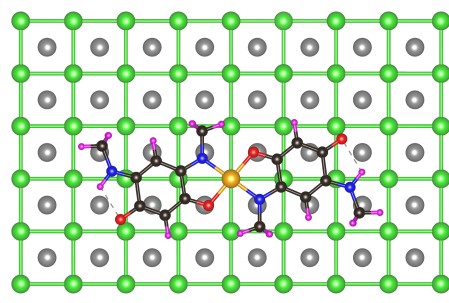

**Figure A1.** (**Left**) DFT+$U$ calculated total energy ($\Delta$E) of NiQ on Co as function of the Ni atom-substrate distance. The curve clearly shows a shallow energy minima at 3.5 Å which corresponds to physisorption of the NiQ molecule on Co(001) substrate in TOP configuration. (**Right**) top view of the physisorbed NiQ molecule on the surface of Co(001), in TOP configuration. Ni, O, N, C, H and Co atoms are represented with orange, red, blue, black, magenta and green colored balls. Second layer (hollow site) Co atoms are represented by grey colored balls.

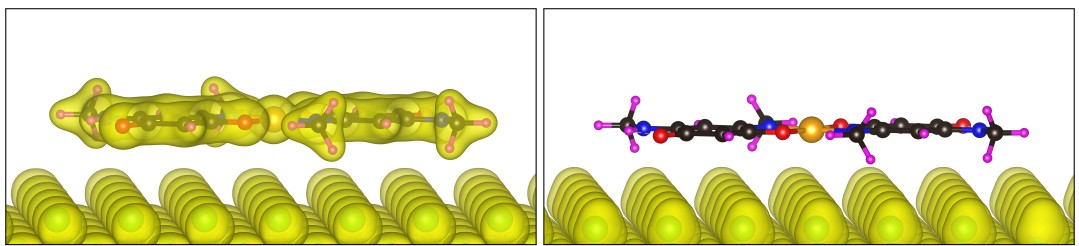

**Figure A2.** Calculated charge density (**left**) and magnetization density (**right**) of physisorbed NiQ molecule on Co(001) in the TOP configuration. The computed magnetization density shows that there is no magnetization density present on the molecule. The bright yellow hypersurface depicts the positive magnetization density.

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
