# Peer review of "Interfacial Spin Manipulation of Nickel-Quinonoid Complex Adsorbed on Co(001) Substrate"

_magnetochemistry, doi:10.3390/magnetochemistry5010002_

Round 1

Reviewer 1 Report

I find the paper by Reddy and coworkers to be of interest. The subject is hot.  It is well done and clearly presented and definitely worthy of publishing. I have one question and one comment, though.
The question is: according to the calculations of the authors, the adsorption of the title complex on the Co surface leads to a very high elongation of the metal-ligand bonds, being ca. 0.26 and 0.15 Ang for Ni-O and Ni-N, respectively. This distances definitely correspond to a high-spin configuration of the complex. Yet, for NiN2O2 coordination the high-spin tetrahedral complexes are possible, revealing the planar-tetrahedral spin equilibria in solution. Would not that make sense that the authors at least calculate the energy of the tetrahedral complex in gas phase?
Secondly, I would suggest to point out that thermodynamically the low-spin to high-spin switching brings about the entropy increase due to electronic and vibrational effects of the spin change. This could be an additional contribution to the driving source of chemisorption.

Reviewer 2 Report

Reddy and co-workers addressed by means of density functional theory (DFT) simulations how the electronic structure of a Ni(II) complex changed upon adsorption on a Co surface. Their results, which are very well presented in the manuscript, demonstrates that the molecule undergoes a spin-state switching and, furthermore, that its spin becomes ferromagnetically coupled to the substrate.

I found the conclusions very interesting and I think that the study represents an important contribution to the understanding and design of novel molecule-metal interfaces for spintronics. I fact, the main predictions may also inspire new surface science experiments as well as the synthesis of other molecular compounds, which are similar to the specific Ni(II) complex discussed in the manuscript. Because of all these motivations, I strongly recommend the publications in Magnetochemistry. Nonetheless I have a few comments and questions I would like the authors to address.

1) The authors state that the calculations are performed by using the DFT+U method with U and J respectively set to 5 eV and 1 eV. Are all the presented results obtained with those values? Why U and J were set to those specific values? Is the spin switching robust against changes of those parameters? Is the molecule in the gas phase still low spin in spite of such large U value?

2) At line 124 the authors state that "the ligands connected to the Ni atoms of the  molecule are impetus to stretch and orient themselves in such a way that the central O atoms of the molecule which were initially in the hollow site moves toward the top site and chemically bond with the surface Co atoms.". However, in Fig. 3, the O atoms appear still in the hollow position. Can the authors clarify that?

3) at line 148 the authors argue that a weak physisorption is observed in the top configuration. However, there is no mention about the use of a functional able to describe van der Waals interactions in the computational details. If the authors used "standard" GGA, the "physisorption" would result from a systematic error of the functional rather than from the proper description of the van der Waals interactions. Can the authors explain how they accounted for the van der Waals interactions? 

Reviewer 3 Report

The computational study by Tarafder and co-workers details on-surface chemisorption induced S=0 to S=1 spin state switching of a charge-neutral Ni(II) complex anchored on a ferromagnetic Co(001) surface. Overall, the study is well conceived and executed. The presentation is clear and concise. Although the results presented in the present study is not new considering the state of the art, publication of this study is recommended in Magnetochemistry due to the novelty associated with the molecular system the authors chose to study.

The following are some minor remarks:

1. The reference citing spin-crossover(SCO) in the introduction is old and not capturing the recent trends. To aid the reader, the authors should consider adding a recent reference covering SCO from the device perspective, for e.g., 10.1016/j.ccr.2017.03.024

2.  What will happen to the spin-state of the molecule if the same molecule is studied on a non-magnetic metallic surface? The quenching/modification of SCO upon interaction with a metallic surface is a standard problem faced by the community and its some of our dreams to realize switchable/bi-stable spinterface by reversibly manipulating the spin-state of a surface anchored molecule. Maybe a perspective, in a few lines, at the end of the script will make the study more meaningful.
